# Effect of Rice Husk-Based Silica on the Friction Properties of High Density Polyethylene Composites

**DOI:** 10.3390/ma15093191

**Published:** 2022-04-28

**Authors:** Yafei Shi, Miaomiao Qian, Xinru Wang, Wanjia Zhang, Xuewei Zhang, Xiaofeng Wang, Yanchao Zhu

**Affiliations:** 1College of Chemistry, Jilin University, Changchun 130012, China; shiyf20@mails.jlu.edu.cn (Y.S.); qianmm19@mails.jlu.edu.cn (M.Q.); xinru21@mails.jlu.edu.cn (X.W.); 18347037230@163.com (W.Z.); xuewei20@mails.jlu.edu.cn (X.Z.); 2State Key Laboratory of Inorganic Synthesis & Preparative Chemistry, College of Chemistry, Jilin University, Changchun 130012, China; wangxf103@jlu.edu.cn

**Keywords:** biomass, rice husk ash, high density polyethylene, friction coefficient

## Abstract

Rice husk ash (RHA)-reinforced composites are now used in many tribological applications. We prepared two kinds of RHAs using different pretreatment and the same pyrolysis process, namely water-treated RHA (WRHA) and acid-treated RHA (ARHA). Comparing the two RHAs, the RHA pretreated with hydrochloric acid (HCl) was found to have a smaller particle size and a more uniform dispersion. Accordingly, the two kinds of RHAs were used as fillers and added to the high-density polyethylene (HDPE) matrix by an extrusion process. The results showed that the friction coefficient (COF) value of the composites with ARHA was reduced to 0.12 when an additional amount of 0.75 wt.% or 1.5 wt.%. WRHA was used as a filler to the amount of 1.5 wt.%, but the COF value was raised to about 0.21. The reason for this phenomenon may be due to its larger particle size and more severe abrasive wear. This work provides a method for making natural biomass fillers that can effectively reduce the COF of HDPE composites with slight decreases in mechanical properties.

## 1. Introduction

Material friction pervades nearly all aspects of our daily lives and plays an important role in life [1]. Due to the influence of friction, the service life of many materials has been greatly reduced. Therefore, fillers are introduced into polymer resin to improve the friction properties of the materials [2].

Nano silica (SiO_2_) is effective in reducing the friction and wear of composites as a filler in poly(ether-ether-ketone) (PEEK) [3]. In addition, physical properties such as shape and size affect the reinforcing properties of the SiO_2_ in the matrix. Studies have shown that the smaller the size of the SiO_2_, the more conducive they are to improvements in the mechanical properties and friction properties of the composites [4]. However, the raw materials silicate and silicon alkoxides that are used to synthesize nano SiO_2_ have high costs [5]. Therefore, it is urgent to find an economical raw material to prepare the SiO_2_. Rice husks (RHs) are composed of 20% silica and 80% lignocellulose by weight, constituting a unique lignocellulose–silica network [6]. Previous studies have shown that SiO_2_ can be prepared from RHs by heat treatment and used as a polymer filler to improve the performance of the composites [7,8]. However, the SiO_2_ produced by this method has low purity, large particles, and non-uniformity [9].

The silica in RHs is mainly concentrated in two places: the protrusions or hairs of the outer epidermis (trichomes) and the inner epidermis (near the grains) [10]. Part of the silica may be combined with polysaccharides to form a bond between silicon and organic compounds [11]. Therefore, many authors recommended refluxing RH with hydrochloric acid (HCl) before the heat treatment to accelerate the hydrolysis of cellulose and hemicellulose and to remove most of the metal impurities [12]. This exposes the silica as much as possible, which is beneficial for RHA with smaller particle sizes. In addition, some studies have shown that the decomposition of organics in RH can effectively improve the heat treatment efficiency, which may reduce the pyrolysis temperature [13]. The results showed that a very low concentration (3 wt.%) of HCl can effectively hydrolyze hemicellulose [14,15,16]. The RHA produced by this method exhibits the characteristics of smaller particle size, higher purity, and higher specific surface area [17,18]. Inspired by these results, we chose 3 wt.% HCl for RH pretreatment and set the heat treatment temperature to 550 °C. As a control in this work, we also heat-treated RH without acid treatment. We also compared RHAs prepared under two different conditions with commercially precipitated nano-SiO_2_. All three silicas are composed of nanospheres, but the commercially precipitated nano-silica has a smaller particle size but more aggregated.

Due to the outstanding chemical stability, environmental stress-crack resistance properties and corrosion resistance of acid, alkali and various salts among polymeric materials, HDPE has been viewed as having great potential for a broad range of applications, such as containers, toys, medical supplies, wires and cables jackets [19]. It is reported that RHA is used as a filler in HDPE to study the mechanical properties and flame retardancy, but it has not been reported in terms of friction properties [20,21,22]. Thus, the two kinds of RHAs were incorporated into HDPE by the melt blending process. We studied the effect of the two kinds of RHAs on the mechanical properties and the friction properties of the composites with different filler amounts. We found that a small amount of RHA, which was produced by refluxing with 3 wt.% HCl before heat treatment as a filler, can effectively reduce the friction coefficient (COF) of the composites with a slight change in mechanical properties. This has a certain significance for the full utilization of biomass-based composite materials and this renewable resource.

## 2. Experimental

### 2.1. Materials

The raw RH was purchased from a mill around Changchun City, China. HDPE was supplied by PetroChina Co., Ltd. (Beijing, China), with a melt flow index (MFI) of 7.6 g/10 min. Commercial precipitated silica was purchased from Shanghai Kaiyin Chemical Co., Ltd. (Shanghai, China), and the product brand is LK-325. HCl in analytical grades was obtained from Sinopharm Chemical Reagent Co., Ltd. (Shanghai, China).

### 2.2. Preparation of SiO_2_ from Rice Husk

RH was washed with distilled water to remove impurities and dried in an oven at 80 °C for 10 h (named as WRH). Then, the WRH sample was refluxed with a 3 wt.% HCl solution (10 mL/g of sample) at 180 °C for 3 h. The products obtained were named ARH. To prepare RHA, the two kinds of RHs (WRH and ARH) obtained above were treated in a high-temperature atmosphere furnace (Shanghai, China) at 550 °C for 3 h under an air flow of 180 L/h. The two kinds of RHAs obtained were named WRHA and ARHA (Table 1), respectively.

### 2.3. Preparation of HDPE-SiO_2_ Composites

HDPE was mixed with the fillers in a co-rotating twin-screw extruder (MINILAB, ThermoHaake, Karlsruhe, Germany). According to the melting points of HDPE, the extrusion temperature was set 180 °C. The screw revolution speed was fixed at 30 rpm. Each sample was extruded three times. The formulation and naming of the prepared composites are shown in Table 2.

### 2.4. Preparation of Test Specimens

The composites were cut into granular samples with a plastic grain-cutting machine. The granular samples obtained were molded into the specific shape by a Mini Jet injection molding machine (SZS-30, Wuhan Ruiming, Wuhan, China) for measurements. The injection temperature was 180 °C, and the mold cooling temperature was 50 °C, with the injection pressure of 0.5 MPa. The sample size of the flexural test was 80 × 10 × 4 mm, and the tensile test was 75 × 10 × 5 mm. The friction sample size was 16 × 10 × 7 mm.

### 2.5. Characterization

The fillers were analyzed by X-ray diffraction (XRD, Empyrean, PANalytical B.V., Almelo, Netherlands) with Cu Kα radiation. The surface characterization of fillers was conducted by Fourier transform infrared spectroscopy (FTIR, Nicolet is5, Bruker, Saarbrucken, Germany). Thermogravimetric analysis (TGA, STA499F3, NETZSCH, Bavaria, Germany) of the fillers and composites was carried out in air flow. The samples were heated from 20 °C to 800 °C at 10 °C/min. The morphology of the filler particles and the HDPE composites were observed using a field-emission scanning electron microscope (FE-SEM, SU8020, Hitachi, Tokyo, Japan). The surface area, pore volume, and average pore diameter of fillers were determined by nitrogen (N_2_) physisorption–desorption at a desorption temperature of 200 °C using a fully automatic multifunctional gas adsorption instrument (ASAP 2020 Plus HD88, Micromeritics, Norcross, GA, USA) and the Brunauer–Emmett–Teller (BET) method. Tensile testing of the HDPE composites was obtained on the universal testing machine (CMT-20, Jinan Liangong, Jinan, China) at a crosshead speed of 10 mm/min, as per ISO 527-1:2012. Flexural testing of the HDPE composites was obtained at a crosshead speed of 2 mm/min, as per ISO 178:2010. The friction coefficient was measured using a Block-on-Ring friction testing at a speed of 200 r/min by the tribometer (CETR UMT-2, Bruker, Saarbrucken, Germany). The sliding time was 2400 s with a load of 45 N.

## 3. Results and Discussion

### 3.1. Characterization of Fillers

The XRD patterns of the three fillers samples are shown in Figure 1a. It can be seen that all broad diffused peaks between 20° and 30° (2 Theta) confirm the formation of the three amorphous samples, and the consistency of the three patterns is particularly important for confirming the composition of the two RHAs. These results are consistent with previous reports, which show that the silica obtained from RH is amorphous in the temperature range of 500–600 °C [23]. Furthermore, the temperature treatment is the major factor affecting the existing forms of the silica in RHA. Figure 1b shows the FTIR spectra of the three filler samples. As can be seen, the fillers’ spectral profiles and relative intensities of the bands are rather similar, indicating a similar structure of the three fillers [24]. The absorption peaks at 470 cm^−1^, 800 cm^−1^, and 1110 cm^−1^ correspond to Si-O-Si swing, symmetric stretching, and asymmetric stretching vibration, respectively. There is a broad peak at 3470 cm^−1^ owing to the adsorbed water and the -OH groups. In addition, the absorption peak at 1647 cm^−1^ is attributed to the H-O-H bonds of the adsorbed water. These results demonstrate that RH produces amorphous silica after heat treatment.

Figure 2 presents the morphology of WRHA and ARHA, which are composed of blocks with a few micrometers to tens of micrometers in diameter. Compared with WRHA (Figure 2a), the particle size of ARHA (Figure 2b) is much smaller because of the pre-treatment of HCl, which can effectively degrade organic matter before treating [25]. The morphologies under greater magnification (Figure 2c,d) show that RHA particles of the silica are agglomerates of the nano-sized particles [26]. In addition, the silica in ARHA (Figure 2b) accumulates more loosely and the particles are relatively small. However, there is a serious agglomeration phenomenon due to its small particle size in commercial SiO_2_ (Appendix A).

Figure 3a shows the N_2_ adsorption–desorption isotherms of the three fillers. In the SiO_2_ sample, substantial absorption occurs when P/P_0_ is greater than 0.9, which is a typical type II behavior [27]. Furthermore, there is an obvious hysteresis loop at P/P_0_ above 0.8, indicating that SiO_2_ is rich in macropores. In the two RHAs, there are two hysteresis loops in the medium and high pressure range (0.4 < P/P_0_ < 0.8), a typical type Ⅳ behavior, indicating that there are a large number of mesopores. The curves in Figure 3b are very consistent with the conclusion of Figure 3a. Table 3 shows the BET surface area, pore volume, and average pore diameter of the three kinds of filler particles. Owing to the HCl treatment of RH, the BET surface area of ARHA (289.91 m^2^/g) is greater than that of WRHA (158.80 m^2^/g) and SiO_2_ (155.78 m^2^/g), as the decrease in particle size resulted in an increase in surface area.

The TGA curves and photographs of the three filler samples in air flow are shown in Figure 4. The curves show an obvious weight loss below 150 °C because of the removal of the absorbed water and gas. The weight loss of the SiO_2_ is less than 0.3% within the range of 200–800 °C due to the volatilization of some crystal water. The weight loss above 350 °C can be ascribed to the combustion of carbon in RHAs. Compared to ARHA and SiO_2_, the carbon residue of WRHA is larger, and the difference can be seen from the color of the photo (Figure 4). This can be interpreted as the reflux of HCl, which can more effectively decompose organics so that the carbon can be fully oxidized during the heat treatment process. In addition, the color of ARHA is very similar to SiO_2_.

### 3.2. Photography of Pure HDPE and HDPE Composites

Figure 5 shows the SEM images of the freeze-fractured cross sections of pure HDPE and its composites. Compared to pure HDPE (Figure 5a), the surfaces of 1.5SiO_2_/PE (Figure 5c) and 1.5WRHA/PE (Figure 5d) are relatively rough. Due to the hydrophilicity of the silica and the hydrophobicity of the matrix, the SiO_2_ aggregates considerably in 1.5SiO_2_/PE. Furthermore, the rough fracture surface with severe crack propagation is observed in 1.5WRHA/PE. From Appendix A, we can also see that large lumps of filler appear on the brittle fracture surface in 5WRHA/PE sample. As can be seen, 1.5ARHA/PE (Figure 5b) exhibits relatively smooth fractured surfaces. ARHA particles are more uniformly dispersed in the HDPE matrix with the filler amount of 1.5 wt.%, where crack propagation and large-scale agglomeration hardly exist. With the amount of ARHA filler continuing to increase (Figure 6), the surface of the composites becomes rougher and rougher. This may be due to the formation of networks of fillers.

### 3.3. Mechanical Properties of Pure HDPE and HDPE Composites

The tensile strength and flexural strength of pure HDPE and its composites are presented in Figure 7. Appendix A shows the mechanical properties of pure HDPE and composites with commercial silica (SiO_2_) as a filler. Due to the incompatibility between the matrix and the fillers [28], the effects of the three fillers on the mechanical properties are not very different [19]. The thermal behaviors of pure HDPE and its composites with ARHA are investigated by TGA in air flow, as shown in Figure 8. Comparing the TGA curves of pure HDPE, the thermal stability of the composites increases with ARHA content. The residual amount in the curve is not very different from the number of fillers added. This proves that this processing method can mix the filler with HDPE well.

### 3.4. Friction Properties of Pure HDPE and HDPE Composites

Figure 9 shows the COF curves of pure HDPE and HDPE composites with different fillers and various ARHA filler loadings. Within the first 300 s, the micro-protrusions on the surface were gradually peeled off and smoothed, and the surfaces tended to match. After a transition period, the COFs finally reached a relatively stable level. In Figure 9a, the effects on the COF are different under the same amount for the three fillers. The addition of commercial SiO_2_ does not cause a significant change in COF, which is very close to that of pure HDPE. Due to its large particle size and poor dispersion, WRHA does not play a positive role in the COF of the composite. Compared with others, the small addition of ARHA significantly reduces the COFs of the composites. Figure 9b shows the COF curves of HDPE composites with different mass contents of ARHA. With continuous increase in the filler content, the aggregation of filler particles becomes more and more serious. Accordingly, the friction process causes more severe abrasive wear, and thus, the corresponding COF value also increases [20,29].

Further support is provided by the SEM of worn surfaces in Figure 10. The arrow direction is the sliding direction in Figure 10. The worn surfaces of the composites are very different at a sliding duration of 2400 s. Broad and deep ploughs can be clearly seen on the surface of the pure HDPE (Figure 10a), and the main cause of this wear is the ploughing and cutting action. Compared with pure HDPE, the worn surface of 1.5ARHA/PE is smoother (Figure 10b). This may be due to the addition of a small amount of ARHA, the COF is lower than that of pure HDPE [30]. However, the COF is higher under the same amount of WRHA, so that WRHA as a filler does not have a positive effect on friction. This can also be further corroborated from the exfoliation of the bulk material in Appendix A. This may be due to the different pretreatment methods from ARHA, resulting in larger particles and poorer dispersion, and then, the COF value is also larger. Furthermore, considerable flaking can be seen on the friction surface of the material [31]. From Figure 10b,d,f, the surface exhibits flaky peeling due to an increase in the filler aggregated micro-protrusions when the amount of filler continues to increase [32]. Therefore, the value of the corresponding COF also increases with the increase in filler content.

## 4. Conclusions

This work shows that 3 wt.% HCl pre-treatment and heat treatment at 550 °C can effectively reduce the particle size of RHA and make the dispersion more uniform. ARHA with smaller particle sizes is more compatible with the HDPE matrix. The RHA obtained under the two conditions as the filler of HDPE did not have a great impact on the mechanical properties. Adding a small amount of ARHA as filler into HDPE effectively reduces the friction COF of the composites. When the additional amount is 0.75 wt.% or 1.5 wt.%, the COFs of the composites are about 30% lower than that of pure HDPE. However, due to the larger particle size of WRHA, when the additional amount is also 1.5 wt.%, the COF value of the composite increases by 40%.

## Figures and Tables

**Figure 1 materials-15-03191-f001:**
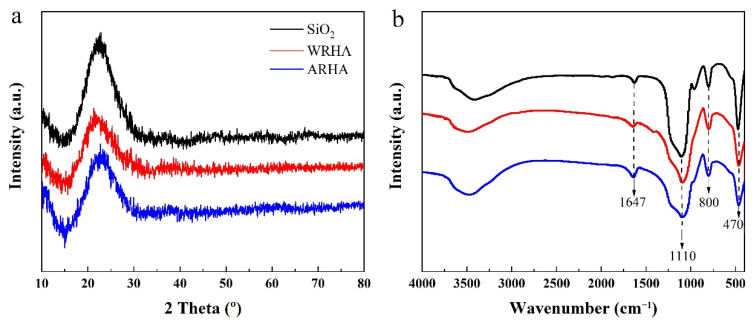
(**a**) XRD patterns and (**b**) FTIR spectra of the three kinds of fillers.

**Figure 2 materials-15-03191-f002:**
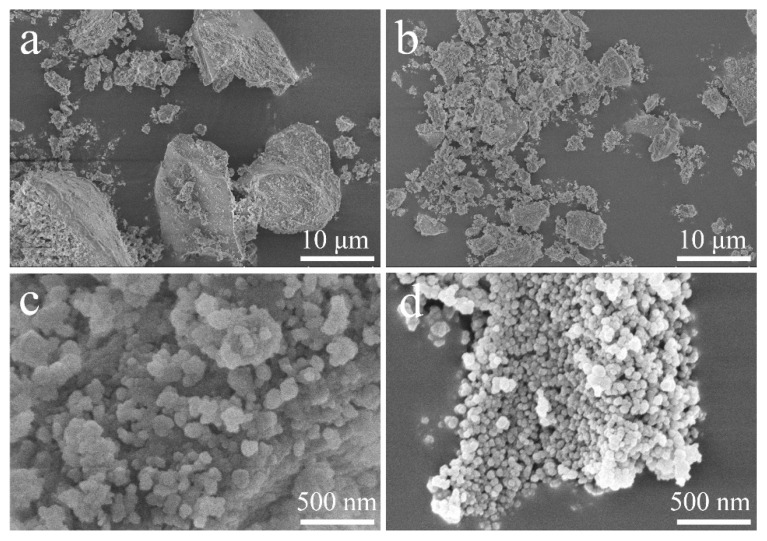
SEM images of WRHA (**a**,**c**) and ARHA (**b**,**d**) at different magnifications.

**Figure 3 materials-15-03191-f003:**
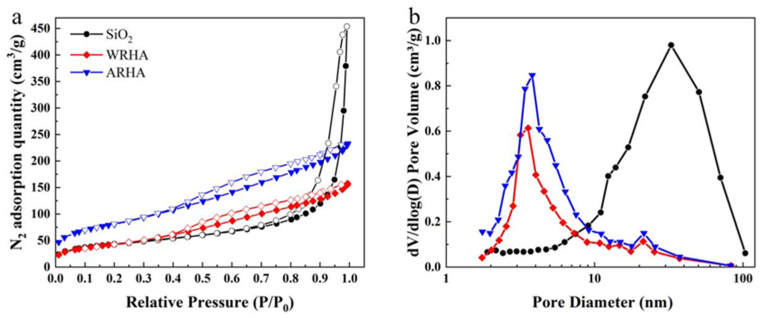
(**a**) N_2_ adsorption–desorption isotherms and (**b**) pore size distributions of the three fillers samples.

**Figure 4 materials-15-03191-f004:**
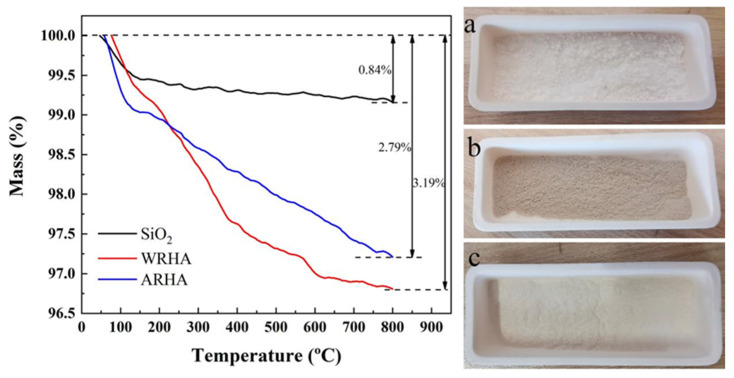
TGA curves in air flow and photographs ((**a**) SiO_2_, (**b**) WRHA and (**c**) ARHA) of the three kinds of filler samples.

**Figure 5 materials-15-03191-f005:**
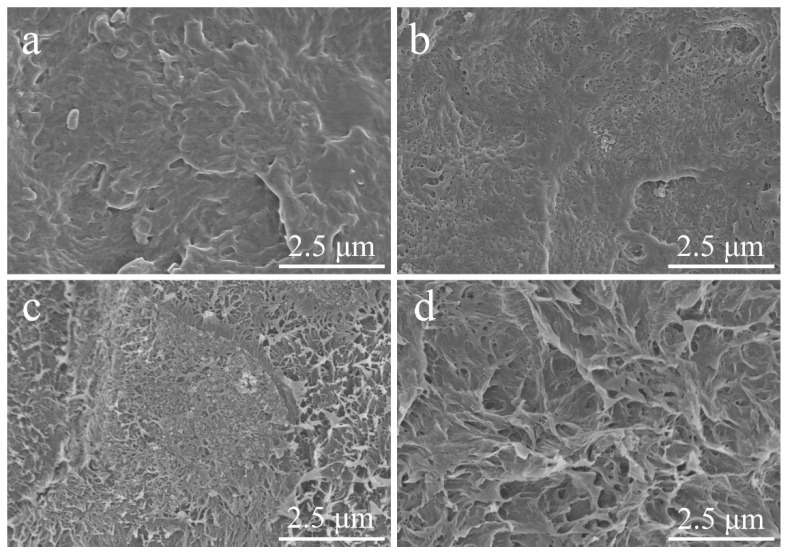
SEM images of (**a**) pure HDPE and its composites brittle fracture surface: (**b**) 1.5ARHA/PE, (**c**) 1.5SiO_2_/PE, and (**d**) 1.5WRHA/PE.

**Figure 6 materials-15-03191-f006:**
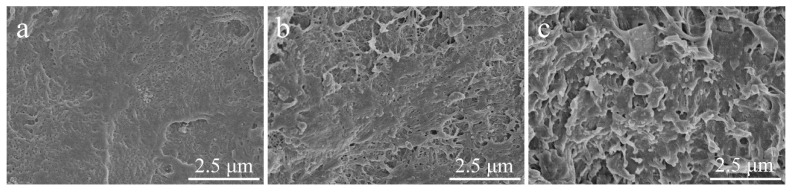
SEM images of HDPE composites’ brittle fracture surfaces: (**a**) 1.5ARHA/PE, (**b**) 5ARHA/PE, and (**c**) 15ARHA/PE.

**Figure 7 materials-15-03191-f007:**
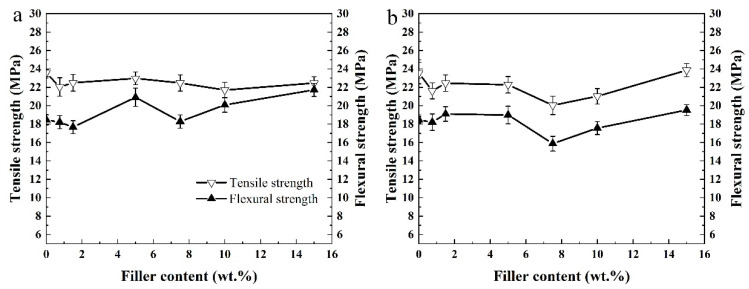
Mechanical properties of pure HDPE and HDPE composites: (**a**) WRHA as fillers and (**b**) ARHA as fillers.

**Figure 8 materials-15-03191-f008:**
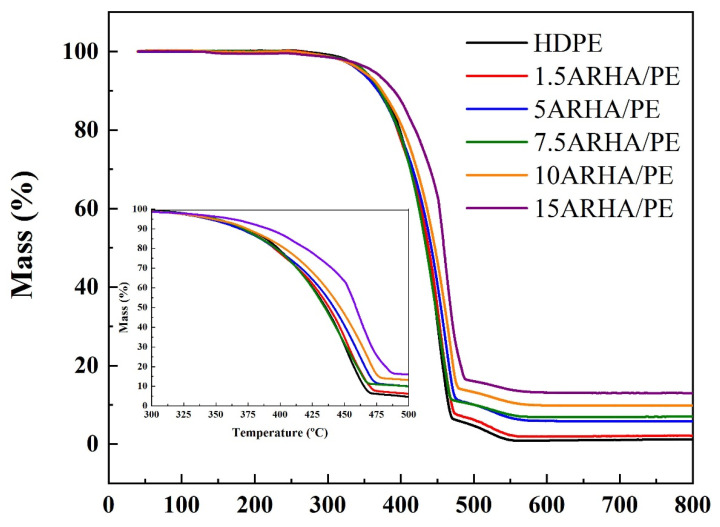
TGA curves in air flow for pure HDPE and its composites.

**Figure 9 materials-15-03191-f009:**
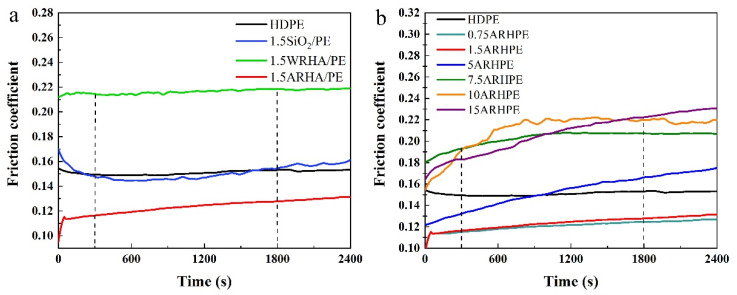
The friction coefficient of (**a**) pure HDPE and its composites with the three kinds of fillers and (**b**) pure HDPE and ARHA/PE composites with various filler loadings with a load of 45 N.

**Figure 10 materials-15-03191-f010:**
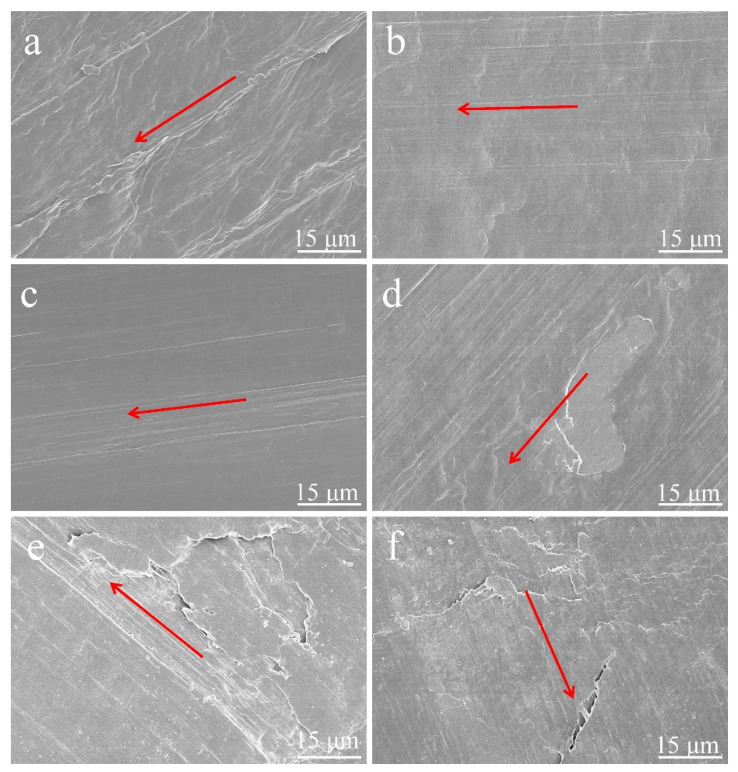
SEM images of worn surfaces of (**a**) pure HDPE and its composites: (**b**) 1.5ARHA/PE, (**d**) 5ARHA/PE, (**f**) 15ARHA/PE, (**c**) 1.5SiO_2_/PE, and (**e**) 1.5WRHA/PE. Arrows indicate sliding direction.The above work proved that nano-sized RHA can be obtained from RH through an appropriate chemical treatment. A small amount of ARHA as a filler in HDPE can reduce the COF of the composites. In addition, the reinforcing effect of ARHA for other polymer matrices is worthy of further exploration.

**Table 1 materials-15-03191-t001:** The production conditions of the two kinds of filler samples.

Samples	Pre-Purification	Temperature (°C)	Air Flow
WRHA	Distilled water	550	180 L/h
ARHA	3 wt.% HCl	550	180 L/h

**Table 2 materials-15-03191-t002:** Composition and names for HDPE composites.

Samples	Composition
XSiO_2_/PE	X g SiO_2_ + (100 − X) g HDPE
XWRHA/PE	X g WRHA + (100 − X) g HDPE
XARHA/PE	X g ARHA + (100 − X) g HDPE

**Table 3 materials-15-03191-t003:** The surface area, pore volume, and average pore diameter of the three fillers.

Samples	BET Surface Area (m^2^/g)	Pore Volume (cm^3^/g)	Average Pore Diameter (nm)
SiO_2_	155.78	0.70	18.02
WRHA	158.80	0.24	6.14
ARHA	289.91	0.36	4.97

## Data Availability

The data presented in this study are available from the corresponding author upon request.

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
