# Peer review of "Effect of Rice Husk-Based Silica on the Friction Properties of High Density Polyethylene Composites"

_materials, 2022, doi:10.3390/ma15093191_

Round 1
Reviewer 1 Report
Yafei Shi et al. obtained silica fillers from rice husk ashes in the present manuscript. To do this, they treated the rice husk with water (WRH) and with hydrochloric acid (ARH). Both treated husks were calcined to obtain the respective rice husk ash (WRHA) and (ARHA).
They extensively characterized WRHA and ARHA and compared them with commercial silica particles. They identified the presence of silica in both WRHA and ARHA. Subsequently, they used different quantities of WRHA and ARHA as fillers to manufacture composites based on high-density polyethylene (HDPE). The authors conclude that the addition of small amounts of ARHA to HDPE reduced the friction coefficient of the composites.
Certainly, Yafei Shi's work has a suitable experimental methodology and is engaging. In this type of experimental work, where an analysis of the influence of a particular element on the physical characteristics of a composite is carried out, it is imperative to carry out an excellent description of both the experimental procedure and its results. Something that the authors DID DO and greatly facilitated the reading of the complete work.
I consider that the work can be published in the Journal Materials if the following adjustments are made (below, I list the corrections requested in the order in which they appeared as I read the manuscript. Obviously, some are more important than others. I ask authors to be much more thorough in those they identify as relevant to improving their work).
- In some parts of the writing of the manuscript, the authors omit the use of necessary articles. For example, in the title, it should be: "Effect of rice husk-based silica on THE friction properties of High-Density Polyethylene composites."
- On line 13, the same thing happens. It should be: Then THE two kinds… (I won't go into more about it, but you should review these minimum details of English throughout the manuscript).
- In the literature, many works talk about the high content of silicon in the rice husk. In fact, the authors do cite some of these works. However, it would be beneficial to delve a little more into the presence of silicon in the rice husk and the components that can be extracted from it, such as silicon nitrides or carbides.
- A classic but important question to answer is why the authors chose 550°C as the combustion temperature. Why not 500, why not 600, why not 700?
- In line 112, it says RHA when she should say ARHA.
- How can the reader be sure that the particles in Figure 1 are silica? At least up to that point in the manuscript, it has not been discussed. When the authors discuss that section, it might be helpful if they put a statement saying "as will be probed later by FTIR."
- The acronym BET on line 141 is not defined.
- In section 3.3, specifically in figure 7, the authors show that they used different compositions of WRHA within HDPE to make composites. However, this is not mentioned in table 2 of the experimental section. The latter is a serious mistake because the reviewer could speculate that the Figure 7 results may not be entirely reliable.
However, it is also possible that the authors have omitted to report those composites in the experimental section. Therefore, and in the understanding that they DO have several composites formed with various amounts of WRHA, the authors are requested to correct the table in the experimental section and that in the manuscript (or in the supporting information document) add and discuss scanning electron microscopy images of said composites. These new images should be similar to those presented in figures 5 and 6, where different amounts of ARHA are used to form HDPE composites. - The authors must improve the captions of figure 9 to describe the figure in-depth so that the reader does not have to consult the text.
Reviewer 2 Report
Revision of “Effect of rice husk-based silica on friction properties of High Density Polyethylene composites”
The manuscript under review devoted to investigate rice husk ash (RHA) reinforced composites that may be used in many tribological applications. Providing of such investigations is very important from an academic point of view (giving new knowledge about the nature of the objects under study) and economic (reducing the cost and improving the quality of the end device).
It was shown that that 3 wt.% HCl pre-treatment and combustion treatment at 550 ºC can effectively reduce the particle size of RHA and make the dispersion more uniform. Acid-treated RHA (ARHA) with smaller particle size is better compatible with the high-density polyethylene (HDPE) matrix. The RHA obtained under the two conditions as the filler of (HDPE) did not have a great impact on the mechanical properties. Adding a small amount of ARHA as filler into HDPE effectively reduces the friction coefficient (COF) of the composites. When the addition amount is 0.75 wt.% or 1.5 wt.%, the COFs of the composites are about 30% lower than that of pure HDPE. However, due to the larger particle size of WRHA, when the addition amount is also 1.5 wt.%, the COF value of the composite increases by 40%.
In manuscript all necessary information is captured by 10 figures, 3 tables and 32 references, all of them are adequate and are reflected in the text.
After getting acquainted with the presented manuscript, a question remained:
- In the text it is necessary to arrange the correct superscripts and subscripts. For example page 1 line 26 "Nano silica (SiO2) is effective in reducing… " or page 4 line 129 “…cm−1, 800 cm−1 and 1110 cm−1 correspond …” etc.
- In the introduction, please provide a physical - chemical justification for the choice of high-density polyethylene as one of the objects.
- Will the results be valid for other polymers?
- It is necessary to check all the designations in the text, since line 112 on page indicates that the results are given for RHA and in the caption for WRHA and ARHA.
- Please, if it possible, explain the step change (almost 2 times) in the modulus shown in fig. 7a.
- In the objects of study, table 2, please add the designations of all the objects under study, since the text contains 0.75ARHA/PE, which is not presented in the specified table.
The obtained results are important both for understanding the physical processes that occur in real objects and for the development of new materials. It corresponds to the field of the Journal «Materials» and may accepted after revision.
Round 2
Reviewer 1 Report
The authors have responded to all the observations of the first round.
This is a beautiful work that I consider should be published in its present form.
Congratulations